# Effects of 220 MHz Pulsed Modulated Radiofrequency Field on the Sperm Quality in Rats

**DOI:** 10.3390/ijerph16071286

**Published:** 2019-04-10

**Authors:** Ling Guo, Jia-Jin Lin, Yi-Zhe Xue, Guang-Zhou An, Jun-Ping Zhang, Ke-Ying Zhang, Wei He, Huan Wang, Wei Li, Gui-Rong Ding

**Affiliations:** 1Department of Radiation Biology, Faculty of Preventive Medicine, Fourth Military Medical University, 169# Chang Le West Road, Xi’an 710032, China; guolingclover@163.com (L.G.); linjiajin913@126.com (J.-J.L.); 15596610582@163.com (Y.-Z.X.); ee26310@126.com (G.-Z.A.); xp10260641@163.com (J.-P.Z.); zhangky@fmmu.edu.cn (K.-Y.Z.); afmmuhw@163.com (W.H.); wanghuan520@xiyi.edu.cn (H.W.); 2Ministry of Education Key Lab of Hazard Assessment and Control in Special Operational Environment, 169# Chang Le West Road, Xi’an 710032, China; 3Department of Histology and Embryology, Fourth Military Medical University, 169# Chang Le West Road, Xi’an 710032, China

**Keywords:** RF field, testis, sperm quality, apoptosis, rat

## Abstract

Under some occupational conditions, workers are inevitably exposed to high-intensity radiofrequency (RF) fields. In this study, we investigated the effects of one-month exposure to a 220 MHz pulsed modulated RF field at the power density of 50 W/m^2^ on the sperm quality in male adult rats. The sperm quality was evaluated by measuring the number, abnormality and survival rate of sperm cells. The morphology of testis was examined by hematoxylin–eosin (HE) staining. The levels of secreting factors by Sertoli cells (SCs) and Leydig cells (LCs) were determined by enzyme linked immunosorbent assay (ELISA). The level of cleaved caspase 3 in the testis was detected by immunofluorescence staining. Finally, the expression levels of the apoptosis-related protein (caspase 3, BAX and BCL2) in the testis were assessed by Western blotting. Compared with the sham group, the sperm quality in the RF group decreased significantly. The levels of secreting factors of SCs and the morphology of the testis showed an obvious change after RF exposure. The level of the secreting factor of LCs decreased significantly after RF exposure. The levels of cleaved caspase 3, caspase 3, and the BAX/BCL2 ratio in the testis increased markedly after RF exposure. These data collectively suggested that under the present experimental conditions, 220 MHz pulsed modulated RF exposure could impair sperm quality in rats, and the disruption of the secreting function of LCs and increased apoptosis of testis cells induced by the RF field might be accounted for by this damaging effect.

## 1. Introduction

In recent decades, male reproductive problems, such as spermatogenesis disorders, infertility, and premature abortion, have become more prevalent worldwide [1]. It is worth noting that infertility affects an estimated 15% of couples globally, and males were believed to contribute to 50% of cases overall [2]. Male reproductive problems have caused wide social concern. Accumulating evidence has shown that radiofrequency (RF) fields, as a kind of nonionizing radiation, potentially has negative effects on human reproductive health [3,4,5,6].

RF fields, a kind of electromagnetic field at the frequency ranging from 100 kHz to 300 GHz, has gone through phenomenal development and deployment worldwide in recent years [7]. With the rapid expansion of devices generating RF fields, the intensity, time and complexity of exposure to RF fields are increasing, which has triggered widespread concern about the safety of RF exposure. It has been reported [8] that the testis was one of the most sensitive target organs to RF fields. Dasdag [9] found that long-term exposure to a 2.4 GHz RF field (testis specific absorption rate (SAR): 0.0024 W/kg, 24 h/day for 12 months) affected some reproductive parameters of male rats, including an increase in abnormities of sperm and the alteration of testis morphology, which could directly or indirectly affect sperm quality.

Currently, 220 MHz pulsed modulated RF fields have been widely used in radio communications. However, there is little research on the bio-effects of 220 MHz pulsed modulated RF exposure, particularly on the male reproductive system. Therefore, the purpose of this study was to investigate the effects of 220 MHz pulsed modulated RF exposure on sperm quality, and to further explore the underlying mechanisms in rats.

## 2. Materials and Methods

### 2.1. Animals

A total of 40 adult male Sprague–Dawley (SD) rats (body weight 237.1 ± 8.1 g), purchased from the Laboratory Animal Center of Fourth Military Medical University (Xi’an, China), were maintained under strict hygienic and well-ventilated conditions at a relative humidity of 50% ± 2%, temperature of 23 ± 2 °C on a 12 h light/dark schedule. Rats lived in standard ventilated polypropylene cages with dry husk as the bedding material, which was changed every two days, and were supplied with standard commercial rodent food pellets and tap water ad libitum. All animal experiments reported in this study were conducted according to the experimental protocol approved by the Research and Education of Fourth Military Medical University (No. 20170607, Xi’an, China), and all efforts were made to minimize animal suffering.

### 2.2. Study Design

A schematic diagram of the exposure system is shown in Figure 1. This system mainly consisted of a signal generator, an amplifier, a coupler, a power sensor and a radiating antenna. The signal generator (N9310A, Keysight Technologies, Santa Rosa, CA, USA) was used to generate the initial carrier signal with preset amplitude in a pulse modulated wave type. The specific signal was amplified via a power amplifier (BLWA 2010-1500, Bonn Elektronik Inc., Ottobrunn, Germany). A stacked Log-periodic antenna (STLP 9128 D special, Schwarzbeck Inc., Schonau, Germany) was used for the exposure in perpendicular polarization. The distance from the antenna to the target animals was 3.5 m, which ensured that the target animals were in the far field of the antenna. A power coupler and an oscilloscope (MSO7054A, Agilent Technologies, Santa Rosa, CA, USA) were combined together to detect the waveform and amplitude of the RF field. The power density was measured in advance by an electromagnetic field meter (PMM8053A, PMM Costruzioni Electtroniche Centro Misure Radio Electriche S.r.l., Milan, Italy). The modulation parameters of the exposed electromagnetic field were 10 μs pulse width and 50% duty cycle.

The rats were randomly divided into the RF group and the sham group (*n* = 20 for each group). The average power density of the 220 MHz pulsed modulated RF field was 50 W/m^2^. The SAR values of whole body and testis were 0.030 W/kg and 0.014 W/kg in the RF group, which were calculated using XFDTD software with a rat model (body mass 307.8 g) provided by Remcom (State College, PA, USA). Animals in the sham group were used to neutralize the experimental box-related and other external constraints and treated in the same way as the RF group but with zero output of the exposure system. All the rats were individually placed in plexiglass experimental boxes, with one rat in each experimental box (26 cm × 8 cm × 7 cm, with small ventilation holes on every side). In each group, the 20 experimental boxes loaded with rats were arranged in a 4 × 5 array and placed on a shelf with the head facing the antenna. Rats were exposed or sham-exposed separately to the above environment in a 220 MHz pulsed modulated RF for 1 h/day (during the light condition). The rectal temperature of all rats were measured immediately before and after 220 MHz pulsed modulated RF exposure, which caused a rise in rectal temperature of <0.5 °C. During exposure, the rats were awake, but not allowed to access food and water, and were kept unrestrained in the plexiglass box. The body weight of rats in the two groups was recorded every three days. The growth curve of body weight and weight gain (weight on 30th day minus weight on 0th exposure day) of the rats in both groups were calculated.

### 2.3. Blood Collection and Tissues Sampling

After one month of 220 MHz pulsed modulated RF exposure, 24 rats were deeply anaesthetized with 1% sodium pentobarbital (60 mg/kg, *n* = 12 for each group). Blood samples were taken from the left ventricle of the heart, and were left at room temperature for at least 30 min, and then the serum was obtained by centrifugation at 3000 rpm for 10 min at 4 °C. Bilateral cauda epididymites and testes were isolated, immediately washed with pre-cooled phosphate buffered saline (PBS, pH 7.4), and dried with filter paper. Cauda epididymites were used to study sperm quality. Bilateral testes were weighed and stored at −80 °C. The testis index was calculated using the following formula: bilateral testes weight (g)/body weight (g) × 100.

The other 16 rats (*n* = 8 for each group) were anaesthetized as previously described and fixed via cardiac perfusion with 4% paraformaldehyde (PFA) after flushing out the red blood cells with 0.9% sodium chloride. The bilateral testes were isolated and fixed in 4% PFA for routine histological examination.

### 2.4. Sperm Quality

Bilateral cauda epididymites were placed in a 12-well plate containing 1 mL sperm culture solution (M2, Millipore, MA, USA), gently cut, shaken slowly for 30 min on a shaker, and were incubated at 37 °C for 20 min. Subsequently, a sperm suspension was obtained. Sperm quality indicators, including the number, abnormality and survival rate of sperm cells, were recorded and calculated accordingly.

The sperm suspension was diluted 10-fold with sperm culture solution and then mixed thoroughly and transferred to a Urine Sediment Quantitative Counting Chamber Slide (Uric-SCP, BMJ Ventures Inc., Surrey, BC, Canada). The number of sperm cells was calculated under a light microscope. This was repeated five times and the results were then averaged (coefficient: × 10^5^/mL). The sperm count was carried out according to the report from Shahin [10].

Sperm suspension was smeared on a slide and allowed to air dry in a dust-free environment. Cells were then fixed by pre-cooled acetone for 20 min, stained with 2% eosin for 1 h, and washed thoroughly with tap water for a few minutes. About 400 sperm cells from different microscopic fields on each slide were examined under the microscope, and the number of abnormal sperm cells was counted according to the standards of abnormal sperm in rats described elsewhere [11].

The survival rate of sperm was assessed by measuring the integrity of the sperm cell membrane. Sperm suspension (20 µL) was added to sperm hypoosmotic medium (200 µL, LEIGEN, Beijing, China) that was pre-heated at 37 °C and smeared on a slide, which was then incubated at 37 °C for 40 min. About 200 sperm (viable and dead) from different microscopic fields on each slide were examined under the microscope. The sperm with an expanding bulge in the tail were recorded as viable, and the survival rate of sperm was calculated using the following formula [10]: survival rate (%) = number of viable spermatozoa/total number of spermatozoa (viable + dead) × 100.

### 2.5. Testes Histology and Seminiferous Tubule Diameter Measurement

Testes fixed in 4% PFA were dehydrated in ethanol, defatted in xylene, embedded in paraffin, and serially sectioned at a thickness of 5 μm on a rotary microtome (LeicaRM2135, Heidelberg, Germany). Subsequently, the sections were stained with hematoxylin–eosin (HE) according to routine procedure. The morphology of the testis was observed using a Leica DMI4000B microscope (Leica, Heidelberg, Germany). The diameter of the randomly selected seminiferous tubules (*n* = 50 for each slide, long axis: short axis < 1.2:1) was measured by the cross method using Image J image analysis software (Image J 1.43u, NIH, Bethesda, MD, USA).

### 2.6. Biochemical Assay Detected by ELISA

Testis tissue (about 150 mg; *n* = 12 for each group) was lysed with radio immunoprecipitation assay (RIPA) lysis buffer, which was supplemented with phenylmethanesulfonyl fluoride and phosphatase/protease inhibitors (cat. No. KGP250, KeyGEN BioTECH, Nanjing, China). The tissue was homogenized to extract total proteins in a homogenate device (Leica, Heidelberg, Germany) under cold conditions. After that, the levels of the secretion factor of Sertoli cells (SCs), such as glial cell line-derived neurotrophic factor (GDNF), stem cell factor (SCF) (SEA043Mu, MEA120Mu, Cloud-Clone Corp., Katy, TX, USA), transferrin (TRF) and androgen binding protein (ABP) (SU-B31079, SU-B35221, Mlbio, Shanghai, China), were measured by a commercial enzyme linked immunosorbent assay (ELISA) kit according to the manufacturers′ instructions. Moreover, the level of serum testosterone (T), which was mainly secreted by Leydig cells (LCs), was determined by an ELISA kit (E-EL-00720, Elabscience, Wuhan, China) using the same method as above.

### 2.7. Western Blot

The concentration of testicular total proteins obtained above was measured using a bicinchoninic acid (BCA) protein assay kit (Beyotime, Nantong, China). About 50 µg of protein from each sample (*n* = 3 for each group) was separated by 12% sodium dodecyl sulfate-polyacrylamide gel electrophoresis (SDS-PAGE) and then transferred onto immuno-blot polyvinylidene fluoride (PVDF) membrane (Millipore, MA, USA). The membrane was then blocked in tris-buffered saline with tween 20 (TBST) buffer containing 5% milk for 2 h at room temperature and incubated overnight at 4 °C with primary antibody using the dilutions listed in Table 1. After 1 h of rewarming the next day, the blots were then incubated with species-matched horseradish peroxidase (HRP)-conjugated secondary antibodies (1:5000, CWBIO, Beijing, China). The bonded proteins were visualized with a chemiluminescent HRP substrate (Millipore, MA, USA) using a Universal Hood II Electrophoresis Imaging Cabinet (Bio-Rad, Milan, Italy), and quantified using Quantity One 4.62 software (Bio-Rad, Milan, Italy).

### 2.8. Immunofluorescence

To confirm the location of apoptosis, immunofluorescence of rabbit monoclonal cleaved caspase 3 (Asp175, CST, MA, USA; dilution 1:400) was carried out on neutral formalin-fixed paraffin-embedded testes sections using the double immuno-labeling method, which was performed in two steps. Briefly, after initial deparaffinization and rehydration of testis section slides, antigen retrieval was performed using citrate buffer in a high-power microwave oven. Testes sections were treated with blocking solution for 2 h at room temperature, then incubated with the aforementioned rabbit monoclonal antibodies overnight in humidity chamber. The following day, sections were washed with PBS and then incubated with goat-anti-rabbit Alexa Flour 594 (Abcam, Cambridge, UK, dilution 1:500) for 2 h at room temperature in the dark. After incubation, sections were again washed with PBS, incubated with 4′,6-diamidino-2-phenylindole (DAPI) for 5 min at room temperature, sealed with nail polish and observed under a fluorescent microscope (Leica, Heidelberg, Germany). The integrated optical density (IOD) of all the areas showing positive signals in the five different regions of the testis sections were measured using Image J 1.43u software and the average IOD values of testes sections were considered as arbitrary unit thresholds (a.u.).

### 2.9. Statistical Analysis

Data were presented as mean ± standard error of the mean (SEM) and graphs were generated using GraphPad Prism 5.04 software (San Diego, CA, USA). Data analysis was performed using Statistical Software Package Statistical analysis system (SPSS 17.0 software, SPSS Inc., Chicago, IL, USA) by the individual blinded to group of exposure. For statistical analysis, repeated measures analysis of variance (ANOVA) was conducted on body weight, and a student *t*-test was used on the other data. *p*-values less than 0.05 or 0.01 were considered statistically significant.

## 3. Results

### 3.1. Effects of RF Exposure on the General Condition of Rats

During a one-month 220 MHz pulsed modulated RF exposure period, no rats were observed to be sick or dead. The body weight of rats in the two groups increased throughout the experiment duration. The body weight (Figure 2A), weight gain (Figure 2B), testis weight (Figure 2C) and testis index (Figure 2D) showed that there were no obvious differences between the two groups (*p* > 0.05).

### 3.2. Effects of RF Exposure on Sperm Quality of Rats

As shown in Figure 3, compared with the sham group, the number of sperm cells decreased markedly and the survival rate of sperm decreased prominently in the RF group (*p* < 0.05). Sperm abnormality increased slightly in the RF group (*p* > 0.05) after 220 MHz pulsed modulated RF exposure. The results suggested that 220 MHz pulsed modulated RF exposure could reduce sperm quality.

### 3.3. Effects of RF Exposure on the Histologic Structure of Rats Testis

Figure 4A shows representative images of testicular morphology under a 10-times and 20-times objective lens, respectively. The layers in the seminiferous tubules of both groups were well organized from external to internal as basal lamina, spermatogonia, spermatocyte, and spermatid. No statistically significant difference was observed in testicular morphology or the diameters of the seminiferous tubules (Figure 4B, *p* > 0.05) between the two groups. 

### 3.4. Effects of RF Exposure on Secreting Function of Rats Testis

As shown in Figure 5, the expression levels of the main secreting factors of SCs, including GDNF, SCF, TRF and ABP, did not change significantly (*p* > 0.05) after one-month 220 MHz pulsed modulated RF exposure, compared with the sham group. This observation suggested that 220 MHz pulsed modulated RF exposure did not affect the secreting function of SCs. However, compared with the sham group, the concentrations of serum T decreased significantly in the RF group (Figure 6, *p* < 0.05) after one-month 220 MHz pulsed modulated RF exposure. The results suggested that the secreting function of LCs could be reduced by 220 MHz pulsed modulated RF exposure.

### 3.5. Effects of RF Exposure on Testicular Cell Apoptosis 

The immunofluorescence of cleaved caspase 3 protein in testes is shown in Figure 7. Compared with the sham group, the expression of cleaved caspase 3 increased markedly in the RF group (*p* < 0.05) after one-month 220 MHz pulsed modulated RF exposure. Cleaved caspase 3 positive cells were mainly distributed in the inner seminiferous tubules. The results of apoptosis-related proteins detected by Western blot (Figure 8) showed that compared with the sham group, the level of caspase 3 protein increased significantly (*p* < 0.01) in the RF group. The level of BAX protein did not change significantly in the RF group (*p* > 0.05), while the level of BCL2 protein decreased significantly (*p* < 0.01) and the BAX/BCL2 ratio increased significantly (*p* < 0.01) in the RF group. These results suggested that 220 MHz pulsed modulated RF exposure could induce testicular cell apoptosis.

## 4. Discussion

According to the report [12] of the Scientific Committee on Emerging and Newly Identified Health Risks (SCENIHR) in 2015 regarding opinions on the reproductive effects of RF exposure, numerous studies have continued to investigate the relationship between RF exposure and male fertility, but none of the recent studies are clearly informative. Most studies have reported that RF exposure could impair sperm quality. Geoffry [13] reported that 1.8 GHz RF exposure in the frequency range of mobile phones and their base station (SAR: 0.4 W/kg–27.5 W/kg, 120 s) could decrease sperm viability and motility of human spermatozoa in vitro. However, negative results have also been found in many studies. A systematic review and meta-analysis [14] around the effect of mobile phones on sperm quality using data from humans and rats showed that mobile phone exposure negatively affects sperm quality, including sperm concentration, motility, viability and morphology. These effects were not consistently reported.

According to current RF exposure standards [15,16], the average power density limit for occupational RF exposure was 10 W/m^2^. In this study, we selected exposure that was five times greater than this limit (50 W/m^2^). We sought to investigate the effects of 220 MHz pulsed modulated RF (a widely used frequency in radio communication) exposure for one month (about two complete spermatogenic cycles; one spermatogenic cycle in rats lasts 13 days [17]) on the sperm quality of adult male rats. After 220 MHz pulsed modulated RF exposure, different sperm quality parameters, the secreting function of SCs and LCs, and the morphology and apoptosis of cells in the testis were observed.

A decrease in the amount and survival rate of sperm cells was found in the RF group after exposing rats to 220 MHz pulsed modulated RF field for one month, which indicated that under this environment, 220 MHz pulsed modulated RF exposure might result in the disruption or impairment of spermatogenesis, which might affect the fertilization capacity of sperm. These findings are consistent with Shahin’s reports [10], who reported that 1800 MHz RF exposure produced by the base station of mobile phones (1.15 W/m^2^, SAR = 0.05 W/kg, 3 h/day for 120 days) decreased sperm count and sperm viability in mice. Meanwhile, Aitken [18] found that after exposure to 900 MHz RF produced by mobile phones (SAR = 0.09 W/kg, 12 h/day for seven days), the sperm count, survival rate of sperm and morphology did not change significantly. Most studies focused on the effect of RF in living environments on sperm quality and the conclusion remains unclear.

Spermatogenesis takes place inside the seminiferous tubules which are composed of SCs and maturing germ cells surrounded by one or more layer(s) of peritubular myoid cells [19]. SCs provide structural and functional support for the development of the maturing germ cells [20] and modulate testis germ cells by expressing several important secreting factors, such as GDNF, SCF, TRF and ABP, thereby creating a local environment optimal for the survival of germ cells. GDNF facilitates communication between SCs and spermatogonia and induces the proliferation of undifferentiated spermatogonia [21,22]. SCF promotes the differentiation of spermatogonia [23]. TRF binds to iron in serum and transports iron to germ cells during development through the blood–testis barrier to maintain normal growth and maturation of germ cells [24]. ABP promotes sperm production and maturation, and provides an appropriate internal environment for spermatogenesis [25]. All the secreting factors play vitally important roles during spermatogenesis. It was found that the levels of the aforementioned growth factors produced by SCs did not alter after 220 MHz pulsed modulated RF exposure compared to the sham group. The available data indicate that under this condition, 220 MHz pulsed modulated RF exposure has no obvious effects on the secreting function of SCs in rats. In addition, body weight, testes weight and the morphology of testis were negligibly affected by 220 MHz pulsed modulated RF field.

Testosterone (T), a steroid hormone, produced mainly by LCs [26], is of vital importance to the development and function of male reproduction, and to the maintenance of normal spermatogenesis [27]. Shahin [10] reported that 1800 MHz RF exposure (1.15 W/m^2^, 3 h/day for 120 days) acted in the male reproductive tract of mice, in part by inhibiting T secretion from LCs. Lin [3] reported that exposing mouse LCs to a 1950 MHz continuous RF wave for 24 h (SAR = 3 W/kg) causes the level of T in the supernatant to diminish dramatically. Similarly, a decrease in the level of serum T was found after RF exposure at a very low SAR value (0.030 W/kg for whole body), which indicated that the effects on T secretion induced by pulsed modulated waves might be more obvious than those induced by continuous waves.

Previous studies have shown that RF exposure under certain conditions could induce apoptosis of testicular cells [28,29]. The protein level of cleaved caspase 3 increased markedly in the RF group compared to the sham group, which was consistent with the result of apoptosis-related proteins in testes detected by Western blot. The above results suggested that under this condition, 220 MHz pulsed modulated RF could induce apoptosis in testis germ cells. Shahin’s report [30] also showed that 2.45 GHz RF (0.248 W/m^2^, 2 h/day for 15 days, 30 days and 60 days) induced testicular apoptosis with an increase of cleaved caspase 3 and BAX, as well as a decrease of BCL2 in a duration dependent manner. Testicular apoptosis may impede germ cell production and result in a reduction of sperm quality.

It was reported that the biological effects of pulse modulated waves were more obvious than those of continuous waves of the same intensity [31]. In this study, we found that 220 MHz pulsed modulated RF at low level SAR impaired sperm quality, which is probably due to the pulse modulated signal. Testes are also known to be a sensitive organ to RF fields [8]. Finally, 220 MHz pulsed modulated RF exposure caused an elevation in rectal temperature of less than 0.5 °C in this study, which indicated that the biological effects of 220 MHz pulsed modulated RF exposure under this condition had nothing to do with thermal effects. The effects of RF on the male reproductive system were contradictory, and all these studies are subject to a variety of methodological limitations, which hampered the further exploration of relative mechanisms.

## 5. Conclusions

These data suggest that under the present experimental conditions, 220 MHz pulsed modulated RF field exposure could reduce sperm quality in rats. The disruption of the secreting function of LCs and the apoptosis in testes induced by RF field might be involved in this process.

## Figures and Tables

**Figure 1 ijerph-16-01286-f001:**
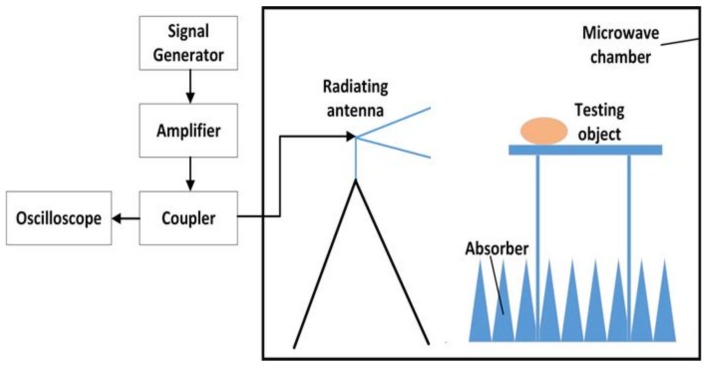
Schematic diagram of experiment set up of the exposure system.

**Figure 2 ijerph-16-01286-f002:**
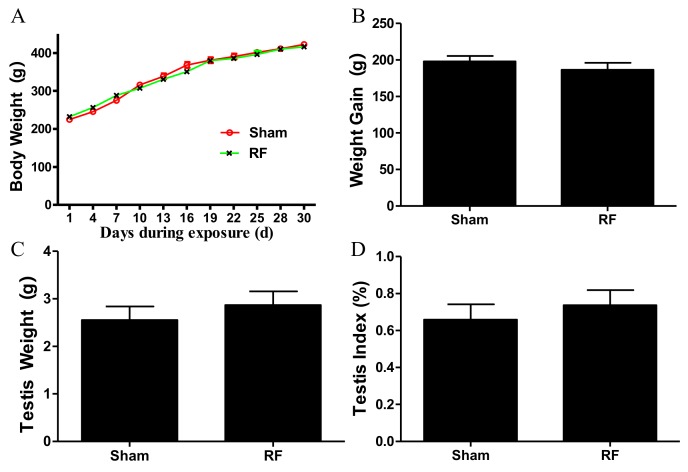
The effect of one-month 220 MHz pulsed modulated radiofrequency (RF) exposure on the general condition of Sprague–Dawley (SD) rats. (**A**) Growth curve of body weight; (**B**) Weight gain; (**C**) Testis weight; (**D**) Testis index (bilateral testes weight (g)/body weight (g) × 100). *N* = 20 for each group. The values were expressed as mean ± standard error of the mean (SEM) and were analyzed by repeated measure ANOVA or student *t*-test.

**Figure 3 ijerph-16-01286-f003:**
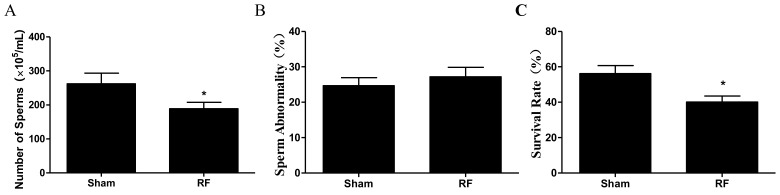
The effect of one-month 220 MHz pulsed modulated RF exposure on sperm quality of SD rats. (**A**) Number of sperm cells; (**B**) Sperm abnormality; (**C**) Survival rate. *N* = 12 for each group. The values were expressed as mean ± SEM and were analyzed by student *t*-test. * *p* < 0.05, compared with the sham group.

**Figure 4 ijerph-16-01286-f004:**
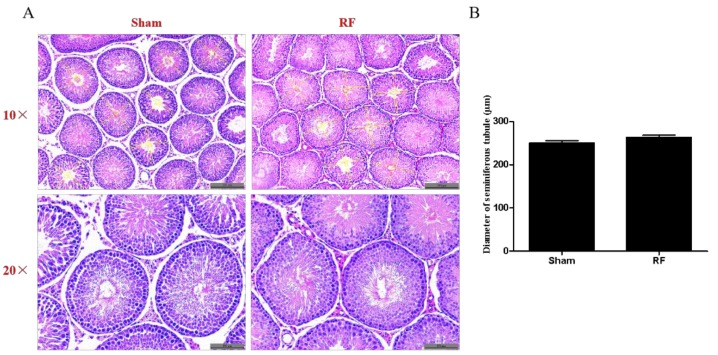
The effect of one-month 220 MHz pulsed modulated RF exposure on testis morphology of SD rats. (**A**) Hematoxylin–eosin (HE) staining, scale bar = 200 μm (upper panel) and 100 μm (lower panel); eight animals for each group. (**B**) Measurement of diameters of seminiferous tubules; 50 seminiferous tubules for each animal and eight animals for each group. The values were expressed as means ± SEM and were analyzed by student *t*-test. Mean differences were no significant (*p* > 0.05).

**Figure 5 ijerph-16-01286-f005:**
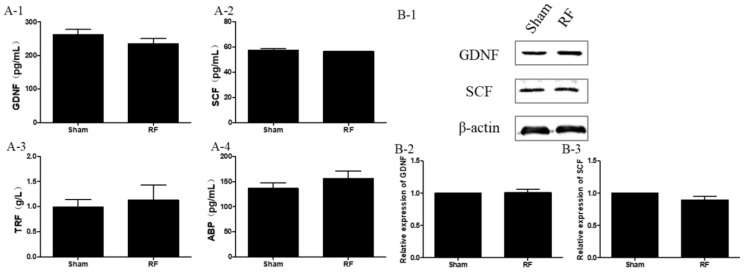
The effect of one-month 220 MHz pulsed modulated RF exposure on the secreting function of testicular Sertoli cells (SCs) in SD rats. (**A**) The results of ELISA. (**A-1**) Glial cell line-derived neurotrophic factor (GDNF); (**A-2**) Stem cell factor (SCF); (**A-3**) Transferrin (TRF); (**A-4**) Androgen binding protein (ABP). *N* = 12 for each group. (**B**-**1**) The results of Western blotting. (**B**-**2**) Relative expression of GDNF; (**B**-**3**) Relative expression of SCF. *N* = 3 for each group. The values were expressed as means ± SEM and were analyzed by student *t*-test. Mean differences were no significant (*p* > 0.05).

**Figure 6 ijerph-16-01286-f006:**
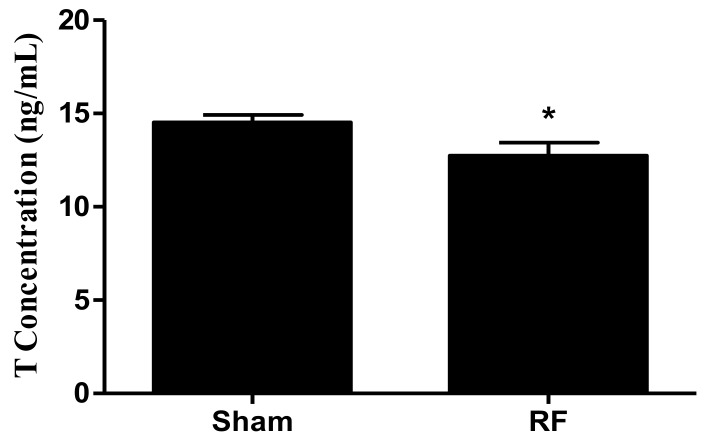
The effect of one-month 220 MHz pulsed modulated RF exposure on the secreting function of testicular Leydig cells (LCs) in SD rats. *N* = 12 for each group. The values were expressed as means ± SEM and were analyzed by student *t*-test. * *p* < 0.05 compared to the sham group.

**Figure 7 ijerph-16-01286-f007:**
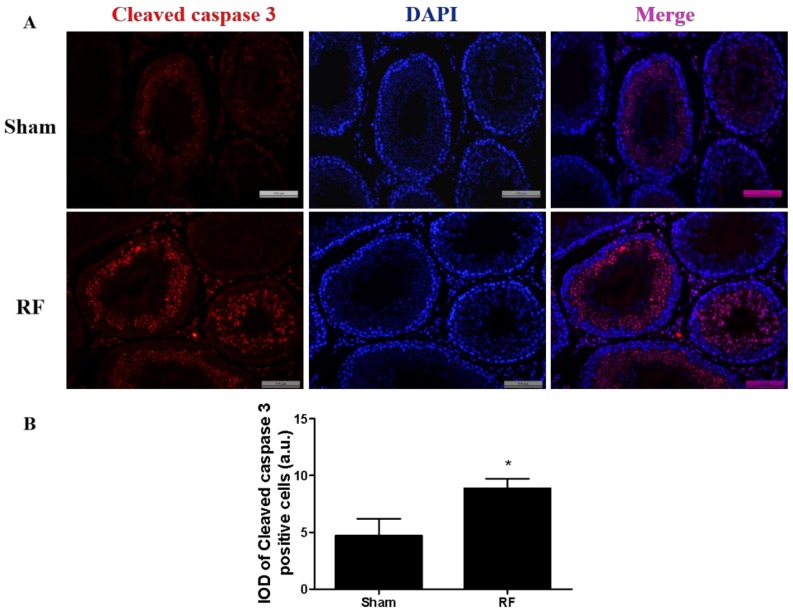
The effect of one-month 220 MHz pulsed modulated RF exposure on cleaved caspase 3 protein in testicular tissue of SD rats. (**A**) Immunofluorescence of cleaved caspase 3, DAPI: 4′,6-diamidino-2-phenylindole; (**B**) the integrated optical density (IOD) of cleaved caspase 3 in the testis. *N* = 3 for each group. The values were expressed as means ± SEM and were analyzed by student *t*-test. * *p* < 0.05 compared with the sham group.

**Figure 8 ijerph-16-01286-f008:**
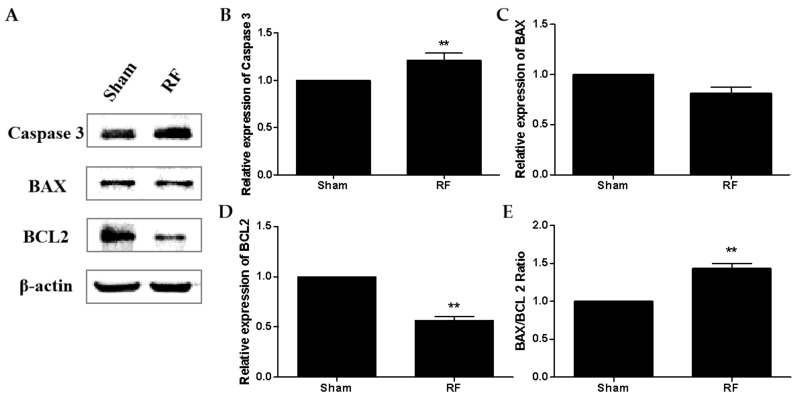
The effect of one-month 220 MHz pulsed modulated RF exposure on the apoptosis-related protein in testicular tissue of SD rats. (**A**) The protein level of caspase 3, BAX, and BCL2 in testicular tissue were measured by Western blot; (**B**) relative expression of caspase 3; (**C**) relative expression of BAX; (**D**) relative expression of BCL2; (**E**) BAX/BCL2 ratio. *N* = 3 for each group. The values were expressed as means ± SEM and were analyzed by student *t*-test. ** *p* < 0.01 compared with the sham group.

**Table 1 ijerph-16-01286-t001:** Antibodies used for Western blot.

Antibody	Species	Company	City and Country	Dilution
Anti-β-actin	Mouse mAb	CMCTAG	Milwaukee, WI, USA	1:5000
Anti-GDNF	Rabbit Polyclonal Ab	Abcam	Cambridge, England	1:400
Anti-SCF	Rabbit Polyclonal Ab	SAB	College Park, Maryland, USA	1:300
Anti-caspase 3	Rabbit Polyclonal Ab	Proteintech	Wuhan, China	1:600
Anti-BAX	Rabbit Polyclonal Ab	Proteintech	Wuhan, China	1:5000
Anti-BCL2	Rabbit Polyclonal Ab	Abcam	Cambridge, England	1:500

Notes: GDNF: glial cell line-derived neurotrophic factor; SCF: stem cell factor; BAX: B cell lymphoma/leukmia-2 associated X protein; BCL2: B cell lymphoma/leukemia 2.

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
