# Peer review of "Effects of 220 MHz Pulsed Modulated Radiofrequency Field on the Sperm Quality in Rats"

_ijerph, 2019, doi:10.3390/ijerph16071286_

Round 1

Reviewer 1 Report

No more questions.

Author Response

Response to comments (Reviewer 1)

All authors would also like to give special thanks to the editor and anonymous referees for their comments and suggestions that greatly improved the substance and presentation of the paper.

Reviewer 2 Report

The manuscript describes interesting experiments of rat’s exposure to radiofrequency field and studies of several parameters characterising the quality of sperm.

The studied problem of potential health effects caused by radiofrequency exposure is very important but some limitations in the description of the performed experiment make difficulties in evaluation of the significance of results presented in the manuscript.

From the previous comments of reviewers and authors’ letters I understand that the description of the biological aspects of experiments was improved. However, in the discussion section authors still not commented how relevant is to discuss published studies, where exposures were 100 times weaker, with respect to results of experiments presented in the manuscript.

But the main problem with the manuscript, in my opinion, is the insufficient description of the technical part of the experiment (what make impossible to replicate it, as well as to compare results with other studies available in the research literature):

1. authors confirmed that exposure was to pulsed modulated RF – in my opinion along the entire manuscript “220 MHz RF” should be replaced by “220 MHz pulsed modulated RF”

2. pulsed modulated field needs to be characterised by the power density over pulse duration and duty cycle ratio – necessary data was not provided – perhaps authors believe that it is confidential military information related to the radar signal used in experiments – but it is well known what frequency bands are used for such purposes – and rough data are not secret at all (however necessary to define exposure conditions during the experiments)

3. in my opinion authors are not able to prove that the uncertainty in the exposure levels were small enough to make possible to distinguish between 50 and 100 W/m2, for example:

a) the parameters of electromagnetic field meter used in controlling exposures were not provided, even the type of measurement device was deleted from the manuscript (section 2.2);

b) authors believe that exposure was to the far field – mentioned in their letter, not in the manuscript – when from the described geometry of exposure it is easy to find, that rats were exposed in the distance from the source equal roughly to the wave length (far field is in larger distance usually);

c) in the manuscript it is no information related to the uncertainty of field levels, as well as uncertainty of SAR estimations, as well as rectal temperature distribution – all of them may help to prove that exposure levels were well controlled, but it means that it is not the case;

d) for SAR calculations it is lack of info about the models of rats used, software, etc., as well as uncertainty of provided numbers – usually it is at the level of +/- (40-50)%;

e) in the authors letter it is mentioned, that they believe in 1dB uncertainty in the exposure level estimation, because of the 12 degree radiation angle – the problem is that 1 dB means roughly +/-25% of uncertainty, and additionally – and a geometry of the exposure set up cause only one component of total uncertainty in the exposure level estimation; next components are caused by uncertainty of the level of emitted power and the differences in exposure of particular rats caused by the reflections and shadow effects from their neighbours – different in the centre of the set of rats’ cages and in the side positions.

Summarising, in my opinion, the experiment performed should not be discussed as the background for the dose-respond in the biological results based on the current content of the manuscript.

The SAR values provided in the 2.2. section seems to be very low with respect to the declared exposure levels, but as long the parameters of pulsed modulation is not provided it is not possible to assess if they are realistic.

In my opinion the publication of this manuscript is reasonable in one of the following cases:

1. along the entire manuscript “220 MHz RF” should be replaced by “220 MHz pulsed modulated RF” and data from the exposed groups are merged [rough analysis based on figures suggest that most of outcomes discussed in the manuscript will still valid – with exception of the comments related to the dose-respond, which should be skipped – when one exposure group is consider only, the level of uncertainty in the exposure level estimation is not very important and may be skipped]

2. along the entire manuscript “220 MHz RF” should be replaced by “220 MHz pulsed modulated RF” and sufficient data related to the exposure parameters (together with uncertainties) should be provided to prove that 2 expose levels were really different.

Author Response

1. According to the comment, “220 MHz RF” was replaced by “220 MHz pulsed modulated RF” in the revised manuscript.

2. According to the comment, some parameters such as pulse width and duty cycle ratio for characterizing pulsed modulated field were provided (P 2, line 37). In addition, the type of measurement device was also added to the revised manuscript (P 2, line 35).

3. According to the comment, some information about far field was added to Materials and Methods section to describe the exposure condition in more detail (P 2, line 31).

4. According to the comment, one exposure group (50 W/m2) was used in the revised manuscript.

5. According to the comment, some information was added to Discussion section to comment the relevance of our data with published studies (P 10, line 14).

This manuscript is a resubmission of an earlier submission. The following is a list of the peer review reports and author responses from that submission.

Round 1

Reviewer 1 Report

The topic of the present manuscript is, in my opinion, interesting and deals with a hot item.

It  aims to investigate the effects  of RF field on the spermatogenesis. To this purpose rats were subjected to one-month exposure (1 h/day) to a 220 MHz RF field at two different average power densities (50 W/m2 and 100 W/m2). After exposure, various parameters were analysed such as the body weight, testis weight and morphology, the number, abnormality and  survival rate of the sperm, levels of factors secreted both  by Sertoli cells ( glial cell line-derived neurotrophic factor (GDNF), stem cell factor (SCF), transferrin (TRF), androgen binding protein (ABP),  and by  Leydig cell (testosterone (T),  and levels of some  apoptosis-related proteins  (Caspase 3, BAX and BCL 2).

The authors found that 220 MHz RF exposure can affect the protein level of caspase 3 and the BAX/BCL 2 ratio in testis tissue; the levels of T in serum, and the sperm quality.

Although being not conclusive, results could be published 

Some minor comments:

Testosterone levels decreased significantly in 50 W/m2 group, and increased significantly in 100 W/m2 group. The authors should discuss these different results.

I suggest to add in each figure the number of rats examined (although explained in the “material and methods” section, it could be better clarified because the number is different for different parameters)

Author Response

Response to Reviewer 1 Comments

Open Review

English language and style

( ) Extensive editing of English language and style required
( ) Moderate English changes required
( ) English language and style are fine/minor spell check required
(x) I don't feel qualified to judge about the English language and style

Yes

Can be improved

Must be improved

Not applicable

Does the introduction provide sufficient background and include all relevant references?

(x)

( )

( )

( )

Is the research design appropriate?

(x)

( )

( )

( )

Are the methods adequately described?

(x)

( )

( )

( )

Are the results clearly presented?

( )

(x)

( )

( )

Are the conclusions supported by the results?

(x)

( )

( )

( )

Response:
 The results were described in more detail.

Comments and Suggestions for Authors

The topic of the present manuscript is, in my opinion, interesting and deals with a hot item.

It aims to investigate the effects of RF field on the spermatogenesis. To this purpose rats were subjected to one-month exposure (1 h/day) to a 220 MHz RF field at two different average power densities (50 W/m2 and 100 W/m2). After exposure, various parameters were analysed such as the body weight, testis weight and morphology, the number, abnormality and  survival rate of the sperm, levels of factors secreted both  by Sertoli cells ( glial cell line-derived neurotrophic factor (GDNF), stem cell factor (SCF), transferrin (TRF), androgen binding protein (ABP), and by Leydig cell (testosterone (T),  and levels of some  apoptosis-related proteins  (Caspase 3, BAX and BCL 2).

The authors found that 220 MHz RF exposure can affect the protein level of caspase 3 and the BAX/BCL 2 ratio in testis tissue; the levels of T in serum, and the sperm quality.

Although being not conclusive, results could be published

Some minor comments:

Testosterone levels decreased significantly in 50 W/m2 group, and increased significantly in 100 W/m2 group. The authors should discuss these different results.

Response: According to the comments, some information was added to the discussion part.

I suggest to add in each figure the number of rats examined (although explained in the “material and methods” section, it could be better clarified because the number is different for different parameters)

Response: According to the comments, the number of rats examined was added to the legends of relevant figures.

Reviewer 2 Report

The work in this study is interesting and potentially informative for people working with RF magnetic fields. However, there are some points that should be addressed before it can be considered for publication. In addition, the authors should get editing help from someone with full professional proficiency in English.  

1.     Why there are no error bars in figure 2A?

2.     Labels in some figures are distorted, or not well aligned, or at different sizes.

3.     As shown in Figure 3, 220 MHz RF exposure could reduce the sperm quality in a dose-dependent manner. What is the basis for choosing these two doses? How about the other doses?

4.     What’s the magnetic field intensity?

5.     The experiments started with 20 mice/group but ended with 12 mice/group. What is the reason for this? What is the standard of choosing 12 mice from 20 mice?

6.     In the method part, “10 rats per cage”, isn’t it too crowed? It may violate the animal welfare regulation.

7.     On page 7, the evidences for apoptosis is very weak. Western blots using PRAP or cleaved caspase 3 as markers, and/or flow cytometry experiments using Annexin V/PI double staining are necessary.

8.     There are several mistakes in the figure legend 5, “ (A-2) TRF and (A-2) ABP” should be “(A-3)TRF and (A-4) ABP”, “(B-1) Relative expression of GDNF, (B-2) Relative expression of SCF” should be “(B-2) Relative expression of GDNF, (B-3) Relative expression of SCF”.

9.     There are many spelling mistakes in the text. For example, on Page 6, line 18, “that” is spelled wrong.

10.  There are many grammatical errors in the text. For example, in the abstract, “a 220 MHz RF fields” should be “a 220 MHz RF field”; “two different average power density” should be “two different average power densities”; on Page 7, line 25, “was” should be changed to “were”.

Author Response

Response to Reviewer 2 Comments

Open Review

English language and style

(x) Extensive editing of English language and style required 
( ) Moderate English changes required 
( ) English language and style are fine/minor spell check required 
() I don't feel qualified to judge about the English language and style 

Response: The English language and style were carefully checked.

Yes

Can be improved

Must be improved

Not applicable

Does the introduction provide sufficient background and include all relevant references?

( )

(x)

( )

( )

Is the research design appropriate?

( )

(x)

( )

( )

Are the methods adequately described?

( )

(x)

( )

( )

Are the results clearly presented?

( )

(x)

( )

( )

Are the conclusions supported by the results?

( )

(x)

( )

( )

Comments and Suggestions for Authors

The work in this study is interesting and potentially informative for people working with RF magnetic fields. However, there are some points that should be addressed before it can be considered for publication. In addition, the authors should get editing help from someone with full professional proficiency in English.

1. Why there are no error bars in figure 2A?

Response: There are error bars in figure 2A, they were just less obvious.

2. Labels in some figures are distorted, or not well aligned, or at different sizes.

Response: According to the comments, all figures were modified. 

3. As shown in Figure 3, 220 MHz RF exposure could reduce the sperm quality in a dose-dependent manner. What is the basis for choosing these two doses? How about the other doses?

Response: According to current RF exposure standards, the average power density limit for RF exposure of occupational was 10 W/m2. Considering the uncertainty of extrapolation from rodent to human, 5 and 10 times greater than the limit, that were the average power density 50 W/m2 and 100 W/m2. At present, little research on the bio-effects of 220 MHz RF exposure were done, particularly on male reproductive system. In this study, we have just investigated the effects of 220 MHz RF at two doses on the sperm quality, much work including the other doses will be further studied.

4. What’s the magnetic field intensity?

Response: The animals in this study were placed in RF far field, other than the near field. Under this circumstance, power density or electric field strength were usually used to describe RF parameters. Based on power density, the electric field strength and magnetic field strength can be calculated. For example, 100 W/m2 power density, the electric field intensity is 195 V/m and the magnetic field intensity is 0.52 A/m.

5. The experiments started with 20 mice/group but ended with 12 mice/group. What is the reason for this? What is the standard of choosing 12 mice from 20 mice?

Response: Randomized 12 rats in each group were used for detecting the quality of sperm and biochemical indexes, and other 8 rats were fixed via cardiac per-fusion with 4% PFA for routine histological examination. This information was described in 2.3. Blood Collection and Tissues Sampling.

6. In the method part, “10 rats per cage”, isn’t it too crowed? It may violate the animal welfare regulation.

Response: We have two kinds of standard ventilated polypropylene cages, small one for mice and big one for rats. In this study, big cages were used, 10 rats per cage, it was not crowed. 

7. On page 7, the evidences for apoptosis is very weak. Western blots using PRAP or cleaved caspase 3 as markers, and/or flow cytometry experiments using Annexin V/PI double staining are necessary.

Response: The protein level of Caspase 3, Bcl-2 and Bax were provided as the evidences for apoptosis, which were usually used for detecting apoptosis. Since there were significant differences in the protein level of caspase 3 and the BAX/BCL 2 ratio between RF group and sham group, other apoptosis makers such as PRAP or cleaved caspase 3 were not done. According to the comment, they will be considered in our future work.

8. There are several mistakes in the figure legend 5, “ (A-2) TRF and (A-2) ABP” should be “(A-3)TRF and (A-4) ABP”, “(B-1) Relative expression of GDNF, (B-2) Relative expression of SCF” should be “(B-2) Relative expression of GDNF, (B-3) Relative expression of SCF”.

Response: The manuscript was modified as suggested

9. There are many spelling mistakes in the text. For example, on Page 6, line 18, “that” is spelled wrong.

Response: These mistakes have been corrected in the manuscript.

10. There are many grammatical errors in the text. For example, in the abstract, “a 220 MHz RF fields” should be “a 220 MHz RF field”; “two different average power density” should be “two different average power densities”; on Page 7, line 25, “was” should be changed to “were”.

Response: These mistakes have been corrected in the manuscript.

Reviewer 3 Report

The manuscript describes interesting experiments of rats’ exposure to radiofrequency field and studies of several parameters characterising the quality of sperm.

The studied problem of potential health effects caused by radiofrequency exposure is very important but some limitations in the description of the performed experiment make difficulties in evaluation of the significance of results presented in the manuscript.

The manuscript presents detailed description of the biological aspects of performed experiments, but some important technical aspects need to be also discussed in the manuscript to support discussed outcome of the study.

There are 2 main aspects of the experiment performed which are not discussed in the manuscript:

1. It is not clear if authors report the outcome from the thermal level of exposure to radiofrequency?

In the manuscript it was mentioned that applied exposure of rats was 5-10-times higher than the limit of occupational exposure (page 7, lines 22-23) – it suggest thermal level of exposure.

It was also mentioned that the rectal temperature was measured and not increased over 0.5C (page 2, lines 46-48) – it suggest non-thermal level of exposure – with respect to the whole body averaged temperature, but even when the whole body averaged electromagnetic energy absorption do not cause the thermal effect, in the extremities the localised temperature increase may happen. The usual way to discuss this problem is to test SAR values in various body sections

Additionally in the page 7 (lines 23-25) it was mentioned “extrapolation from rodent to human” – but not defined what kind of extrapolation is discussed (description and references are welcome very much).

It is a strong need that authors provide comments to the above mentioned issues, and define if reported effects of rats’ exposure may be considered as non-thermal, or thermal effects.

2. The characteristic of rats exposure provided in the manuscript is not precise enough to show the evidence that it is possible to show statistically significant difference between 2 exposure levels applied. Please consider the following:

the electromagnetic field emitted from antenna is not uniform in space – please provide electric field strength and magnetic field strength spatial distribution in the location of exposed rat’s cages

it is not clear if all rats from particular groups (20 rats) were exposed in the same time – in 3 exposure set ups – or one by one in the same exposure set up – the only information provided about is that exposure was 1 h/day (9-10 am) – page 2 lines 41-42

in the case of exposure of all animals together – important is the distance between particular cages, and if each day rats were located in the same cage, or different – because exposure in the cage in the centre may be different than in the side cages

exposure levels were reported by the power density, however it is usually used when exposure is to the far field (in the distance of several wave lengths from the antenna) or in the waveguide – the scheme of experiment shown at figure 1 suggest the exposure near the antenna (wavelength for 220 MHz is approximately 1.4 m) where exposure needs to be characterised by electric (E, V/m) and magnetic (H, A/m) components – at each exposure point evaluated individually  – because it is a space of near field where E and H components varied in space significantly and may have significantly varied E/H ratio (different than 377 ohms characterising far field exposure condition)

it also needs to be discussed the uncertainty of measurement of exposure level – what measurement probes of PMM8053A were used (the frequency bands, the uncertainty of measurement) – in this context it is very important to provide the properties of pulsed modulation of used exposure (for example characterised by the pulse duration/whole cycle duration ratio, i.e.  duty cycle ratio). This parameter is important when the biological outcome is discussed, but it has also key significance when the uncertainty of electric and magnetic field measurement is discussed [consider that the uncertainty of calibration of measurement probe is reported with respect to the continuous wave measurement / calibration – in case of pulsed modulated field measurement the additional uncertainty component appears, and is duty cycle dependent, as well as exposure level dependent); it needs also attention that PMM measurement devices are equipped in the measurement probes containing electric field or magnetic field sensors – when the monitor is indicating power density (W/m2) it is power density calculated based on the electric field measurement result and 377 ohms impedance of far field – it is not properly applicable in the case of near field (near the antenna, power density at each measurement point needs to be evaluated individually as the measured E-field multiplied by the measured H-field.

It is a strong need that authors provide comments to the above mentioned issues.

It seems to be quite probable that when uncertainty of the exposure level estimation is applied to the characteristic of animals’ exposure – both exposed groups will be homogenised (when the difference in exposure is comparable to the uncertainty of its estimation), and considered for example as exposed at the level of 75 W/m2 +/-40%. Please make careful considerations of the exposure characteristics.

But, any of conclusions regarding the exposure characteristic of both groups of rats is developed – it needs more attention the results of analysis of T concentration (fig. 6) – what is the reason that observed changes are non-monotonic with respect to the exposure level?

And minor suggestion – please provide information what part of the rats’ day was taken for exposure – during dark or light condition; and in the description of the statistical methods – it should be mentioned that 2 levels of significance was considered – 0.05 and 0.01.

Author Response

Response to Reviewer 3 Comments

Open Review

English language and style

( ) Extensive editing of English language and style required
( ) Moderate English changes required
(x) English language and style are fine/minor spell check required
( ) I don't feel qualified to judge about the English language and style

Yes

Can be improved

Must be improved

Not applicable

Does the introduction provide sufficient background and include all relevant references?

(x)

()

( )

( )

Is the research design appropriate?

( )

(x)

( )

( )

Are the methods adequately described?

( )

( )

(x)

( )

Are the results clearly presented?

( )

( )

(x)

( )

Are the conclusions supported by the results?

(x)

( )

( )

( )

Response: Spell check was done and the manuscript was improved.

Comments and Suggestions for Authors

The manuscript describes interesting experiments of rats’ exposure to radiofrequency field and studies of several parameters characterising the quality of sperm.

The studied problem of potential health effects caused by radiofrequency exposure is very important but some limitations in the description of the performed experiment make difficulties in evaluation of the significance of results presented in the manuscript.

The manuscript presents detailed description of the biological aspects of performed experiments, but some important technical aspects need to be also discussed in the manuscript to support discussed outcome of the study.

There are 2 main aspects of the experiment performed which are not discussed in the manuscript:

1. It is not clear if authors report the outcome from the thermal level of exposure to radiofrequency?

In the manuscript it was mentioned that applied exposure of rats was 5-10-times higher than the limit of occupational exposure (page 7, lines 22-23) – it suggest thermal level of exposure.

It was also mentioned that the rectal temperature was measured and not increased over 0.5C (page 2, lines 46-48) – it suggest non-thermal level of exposure – with respect to the whole body averaged temperature, but even when the whole body averaged electromagnetic energy absorption do not cause the thermal effect, in the extremities the localised temperature increase may happen. The usual way to discuss this problem is to test SAR values in various body sections

Additionally in the page 7 (lines 23-25) it was mentioned “extrapolation from rodent to human” – but not defined what kind of extrapolation is discussed (description and references are welcome very much).

It is a strong need that authors provide comments to the above mentioned issues, and define if reported effects of rats’ exposure may be considered as non-thermal, or thermal effects.

Response: Thanks for the reviewer’s comments. There were two points were questioned, one was thermal or non-thermal effects, the other was extrapolation, which were also our research concerns.

Firstly, thermal or non-thermal effects. As the reviewer suggested, this problem should be based on SAR values. Because of the shortcomings of the accuracy of animal models, the calculation of local SAR values in some key parts, such as testis and brain, is under research. Therefore, average power density rather than SAR value was shown in this manuscript. We chose 10 and 5 times the MPE value of occupational exposure in the IEEE standard for exposure. We were not sure whether this value was thermal level of exposure in rats. However, according to the results of rectal temperature in the experiment, the applied exposure in this study was not the whole-body thermal level of 220 MHz RF exposure for rats. Whether it is local thermal level of exposure, it needs to be further verified in the calculation or test of SAR value.

Secondly, extrapolation considers the use of SAR values from experimental animals to humans. Because of the difficulties in SAR calculation, especially in local SAR (key parts such as testicles), this work has not been carried out. And this is also one of our future work.

2. Response: In our experiment, the Stacked Log-periodic antenna (STLP 9128 D special) was used to emit RF. The main parameters of the antenna could refer to the product official website @http://www.schwarzbeck.de/en/antennas/logarithmic-periodic-broadband-antennas/stacked-logarithmic-periodic-broadband-antennas.html. The design of our experiment referenced the product datasheet. The limit distance of near - far field is 2.5 m at 220 MHz. Therefore, rats in this experiment was set at the place more than the distance of 2.5 m, which ensures that the exposure is to the far field but not the near field. This is a very important issue for our experiment design. Under this circumstance, the use of power density is most appropriate. The following is our consideration of the questions raised by the reviewer.

Firstly, the electromagnetic field emitted from antenna is not uniform in space–please provide electric field strength and magnetic field strength spatial distribution in the location of exposed rats cages.

As the animals were put in the far field, the electromagnetic field emitted from antenna at the animals is a plane wave. Through reasonable plane layout, the maximum radiation angle relative to antenna central was set less than 12°. Therefore, the maximum difference of the power density at different location of exposed rats cages was less than 1dB.

Secondly, it is not clear if all rats from particular groups (20 rats) were exposed in the same time–in 3 exposure set ups – or one by one in the same exposure set up – the only information provided about is that exposure was 1 h/day (9-10 am) – page 2 lines 41-42.

The animals were divided into three exposure groups (50 W/m2 group, 100 W/m2 group and sham group). Three groups were exposed one by one in the same exposure set up. Therefore, the total time is three hours not just one hour. The time (9-10 am) was a mistake, and this mistake was corrected in the revised manuscript.

Thirdly, in the case of exposure of all animals together–important is the distance between particular cages, and if each day rats were located in the same cage, or different – because exposure in the cage in the centre may be different than in the side cages. 

 Each day, the rats in exposure groups were individually placed in a plexiglass box, and then was placed on the shelf in the microwave chamber randomly.

Fourthly, exposure levels were reported by the power density, however it is usually used when exposure is to the far field (in the distance of several wave lengths from the antenna) or in the waveguide–the scheme of experiment shown at figure 1 suggest the exposure near the antenna (wavelength for 220 MHz is approximately 1.4 m) where exposure needs to be characterised by electric (E, V/m) and magnetic (H, A/m) components – at each exposure point evaluated individually–because it is a space of near field where E and H components varied in space significantly and may have significantly varied E/H ratio (different than 377 ohms characterising far field exposure condition).

As we discussed above, rats in this study was set at the place which ensures that the exposure is to the far field but not the near field. Under this circumstance, the use of power density is most appropriate.

Last, it also needs to be discussed the uncertainty of measurement of exposure level – what measurement probes of PMM8053A were used (the frequency bands, the uncertainty of measurement)–in this context it is very important to provide the properties of pulsed modulation of used exposure (for example characterised by the pulse duration/whole cycle duration ratio, i.e. duty cycle ratio). This parameter is important when the biological outcome is discussed, but it has also key significance when the uncertainty of electric and magnetic field measurement is discussed [consider that the uncertainty of calibration of measurement probe is reported with respect to the continuous wave measurement / calibration – in case of pulsed modulated field measurement the additional uncertainty component appears, and is duty cycle dependent, as well as exposure level dependent); it needs also attention that PMM measurement devices are equipped in the measurement probes containing electric field or magnetic field sensors–when the monitor is indicating power density (W/m2) it is power density calculated based on the electric field measurement result and 377 ohms impedance of far field–it is not properly applicable in the case of near field (near the antenna, power density at each measurement point needs to be evaluated individually as the measured E-field multiplied by the measured H-field.

Two ways were combined to measure the exposure level. The first is through field strength analyzer of PMM8053A. The second is through the calculation of instrument parameters. The first way is suitable for continuous wave but not the modulation wave. The two methods were in good agreement in our experiments. Moreover, our RF wave in the experiment was continuous wave but not the modulation wave.

Moreover, in this study we mainly focused on the bio-effect of RF field, and the field distribution was carefully designed and calculated before the experiment.

It seems to be quite probable that when uncertainty of the exposure level estimation is applied to the characteristic of animals’ exposure – both exposed groups will be homogenised (when the difference in exposure is comparable to the uncertainty of its estimation), and considered for example as exposed at the level of 75 W/m2 +/-40%. Please make careful considerations of the exposure characteristics.

Response: Thanks for review’s suggestion. Besides, In our experiment, the average exposure value of each group was about 100 W/m2 and 50 W/m2, and the difference between the maximum and minimum exposure dose in each group was less than 1dB.

But, any of conclusions regarding the exposure characteristic of both groups of rats is developed – it needs more attention the results of analysis of T concentration (fig. 6) – what is the reason that observed changes are non-monotonic with respect to the exposure level?

Response: Thanks for review’s comment, which is also our concern. With respect to the results of T concentration, we speculated that lower dose (50 W/m2) RF could inhibit and higher dose (100 W/m2) could promote the secretion of T in mice, to confirm this speculation, further work is needed

And minor suggestion – please provide information what part of the rats’ day was taken for exposure – during dark or light condition; and in the description of the statistical methods – it should be mentioned that 2 levels of significance was considered – 0.05 and 0.01.

Response: According to the comments, some information was added to the manuscript (page 2 line 35; page 4 line 34).

Round 2

Reviewer 2 Report

1.     The rat cage size should be described. According to the international animal guidelines, ADULT rats cannot be housed in one single cage in group of 10. Without rat cage size, I am not sure whether the average room per rat is enough for these adult rats. 

2.     The evidence for apoptosis is not convincing at all. Western blots using PRAP or cleaved caspase 3 as markers, and/or flow cytometry experiments using Annexin V/PI double staining are necessary for ANY scientific paper that wants to prove the involvement of apoptosis.